# Carbonaceous Filler Type and Content Dependence of the Physical-Chemical and Electromechanical Properties of Thermoplastic Elastomer Polymer Composites

**DOI:** 10.3390/ma12091405

**Published:** 2019-04-30

**Authors:** Jose Ramon Dios, Clara García-Astrain, Pedro Costa, Júlio César Viana, Senentxu Lanceros-Méndez

**Affiliations:** 1GAIKER, Parque Tecnológico, Ed 202, 48170 Zamudio, Spain; dios@gaiker.es; 2BCMaterials, Basque Center for Materials, Applications and Nanostructures, UPV/EHU Science Park, 48940 Leioa, Spain; claragarciaastrain@gmail.com; 3Center of Physics, University of Minho, Campus de Gualtar, 4710-057 Braga, Portugal; 4Institute for Polymers and Composites (IPC), University of Minho, 4800-058 Guimarães, Portugal; jcv@dep.uminho.pt; 5IKERBASQUE, Basque Foundation for Science, 48013 Bilbao, Spain

**Keywords:** polymer composites, nanocarbonanceous fillers, thermal annealing, piezoresistive materials

## Abstract

Graphene, carbon nanotubes (CNT), and carbon nanofibers (CNF) are the most studied nanocarbonaceous fillers for polymer-based composite fabrication due to their excellent overall properties. The combination of thermoplastic elastomers with excellent mechanical properties (e.g., styrene-b-(ethylene-co-butylene)-b-styrene (SEBS)) and conductive nanofillers such as those mentioned previously opens the way to the preparation of multifunctional materials for large-strain (up to 10% or even above) sensor applications. This work reports on the influence of different nanofillers (CNT, CNF, and graphene) on the properties of a SEBS matrix. It is shown that the overall properties of the composites depend on filler type and content, with special influence on the electrical properties. CNT/SEBS composites presented a percolation threshold near 1 wt.% filler content, whereas CNF and graphene-based composites showed a percolation threshold above 5 wt.%. Maximum strain remained similar for most filler types and contents, except for the largest filler contents (1 wt.% or more) in graphene (G)/SEBS composites, showing a reduction from 600% for SEBS to 150% for 5G/SEBS. Electromechanical properties of CNT/SEBS composite for strains up to 10% showed a gauge factor (GF) varying from 2 to 2.5 for different applied strains. The electrical conductivity of the G and CNF composites at up to 5 wt.% filler content was not suitable for the development of piezoresistive sensing materials. We performed thermal ageing at 120 °C for 1, 24, and 72 h for SEBS and its composites with 5 wt.% nanofiller content in order to evaluate the stability of the material properties for high-temperature applications. The mechanical, thermal, and chemical properties of SEBS and the composites were identical to those of pristine composites, but the electrical conductivity decreased by near one order of magnitude and the GF decreased to values between 0.5 and 1 in aged CNT/SEBS composites. Thus, the materials can still be used as large-deformation sensors, but the reduction of both electrical and electromechanical response has to be considered.

## 1. Introduction

Polymers and polymer composites have been receiving increasing interest as promising materials for a large number of applications due to their low cost, simple processing, and lightweight character, and particularly for the large number of possible filler and polymer combinations enabling a vast array of applications. Beyond their reinforcement role, fillers can provide or modify specific properties of the polymer, particularly when using nanofillers that allow the degree of crystallinity [1,2] and the mechanical [3,4,5], electrical [5,6], or thermal [7] properties of the polymer to be tuned. Thus, polymer-based composite materials are on the rise as a scientific and technological field [8].

The multifunctionality of polymer-based composites can be further explored with additive manufacturing processing technologies [9]. Polymer-based multifunctional materials have typically been processed by solvent or extrusion methods, each targeting specific applications. Thus, whereas solvent-casting processing is appropriate for screen, spray, or inkjet printing, extrusion is appropriate for the development of filaments for 3D printing. Polymer extrusion is a solvent-free and scalable process that is widely used at the industrial level [10], whereas solvent-based methods result in a more homogeneous filler dispersion with fewer and smaller agglomerates [10,11]. The mechanical, electrical, or thermal properties of the composites are closely related to the processing method. Due to geometry and nanofiller agglomerations, composites with nanocarbonaceous fillers need well-controlled manufacturing processes in order to obtain composites with homogeneous and uniform overall properties [12]. Homogeneous nanofiller distribution and dispersion within the polymer matrices also depend on the fillers (van der Waals interactions and geometry) [6,13,14], polymer type [13,15], or surfactant [13,16] incorporation into the composite, as well as on the processing method, as already discussed [13,15].

The number of carbonaceous-nanofiller-reinforced polymeric materials has increased within the past decade in view of the functionality and applicability of these composites in a wide variety of applications and devices [2]. The dimensional features of the carbon nanofillers, such as the surface area of graphene (G), the larger aspect ratio and superior mechanical properties of carbon nanofibers (CNF), or the electrical and mechanical properties of carbon nanotubes (CNT), make these nanomaterials interesting for use in the development of multifunctional composite materials [8]. Besides mechanical reinforcement [5], carbonaceous nanofillers with tailored electrical properties can be used to develop piezoresistive composites for force and deformation sensing [9]. 

Among the most common carbon allotropes, graphene is a two-dimensional material with excellent electrical conductivity, strong mechanical properties, and small size, on the order of a few nanometers [12]. There are several research works using polymer composites with graphene showing the applicability of the material for functional applications with a wide range of polymer matrices [12]. In the same way, CNF prepared by vapor growth processes are an important filler to impart mechanical reinforcement and structural health monitoring (SHM) sensing capabilities [12]. CNF were among the first-used carbon nanofillers with high aspect ratio and improved mechanical and electrical properties. CNT with tubular structure, with single- or multi-walled sheets rolled upon themselves, also show excellent mechanical, electrical, and chemical properties [12], are the most-studied carbonaceous filler, and are used in a wide range of applications with varying polymers. [12].

The low percolation threshold obtained in polymer-based composites with carbonaceous nanofillers increases their applicability, whereas the main mechanical properties of the polymer matrices remain intact for low filler contents [12,17]. Thus, CNF, CNT, and graphene are ideal materials for the design of multifunctional polymer composites, from fundamental research to device applications. 

Among the different polymer families, thermoplastic elastomers (TPEs) and elastomers present specific mechanical properties. Their larger deformation and easy recovery capabilities combined with their low cost and ease of processing make these materials appropriate for soft applications such as robotics, artificial skin and muscles, pressure sensors, SHM sensors, and biomedical devices [17,18]. TPEs combine the mechanical properties of elastomers with the chemical stability and processability of thermoplastics, without the need of vulcanization steps [18]. Triblock copolymers such as styrene-b-(ethylene-co-butylene)-b-styrene (SEBS) show exceptional properties, aiming at a variety of applications, such as sensors and actuators [17], biomaterials such as smart skin [19], or printable sensors [20]. The SEBS copolymer is composed of soft (ethylene-butylene) and hard (styrene) domains. The soft blocks are anchored on both sides by the hard blocks, and the material can be processed in various chain morphologies, depending on the molecular weight of the hard and soft domains and their architecture [19]. The SEBS properties can be tailored by controlling the soft/hard ratio or the domains’ architecture (e.g., linear or star). Further, due to the hydrogenation of the butadiene block, SEBS is a biocompatible TPE approved by the FDA. Moreover, apart from its biocompatibility, the hydrogenation of SEBS also imparts resistance to UV radiation and thermal degradation to the polymer [18]. Usually, carbonaceous nanofillers dispersed in TPE matrixes have similar filler distribution within both styrene and ethylene/butylene polymeric domains [19], and the nanofillers do not influence the microstructure of the TPE block copolymer [19]. 

Thermal treatments of TPE matrices can modify their nano- and microstructure [21], as well as their thermal and mechanical properties [21]. As an example, thermal annealing can promote the interdigitation of the SEBS chains, the development of microphase separation, as well as styrene restructuration segments [19]. 

The main purpose of the present work was to evaluate the effects of different carbonaceous nanofillers in SEBS matrices in terms of thermal, mechanical, electrical, and piezoresistive properties. Further, thermal annealing was performed to evaluate the eventual modifications of the material characteristics after reaching temperatures up to 120 °C. 

## 2. Materials and Methods

### 2.1. Materials

The polymer used to produce the composites was a thermoplastic elastomer (TPE) composed of styrene-b-(ethylene-co-butylene)-b-styrene (SEBS) from Dynasol Elastomers (Madrid, Spain) with reference Calprene H6110 with a 70/30 ethylene-butylene/styrene ratio, a linear structure, polymerized in solution, excellent ozone resistance, a density of 0.91 g/cm^3^, and a hardness of 76 (Shore A). Calprene H6110 meets the requirements of the USP class VI plastic classification.

The reinforcement materials were electrically conductive carbonaceous nanofillers, namely: carbon nanofibers (CNF), carbon nanotubes (CNT), and graphene (G). CNF were from Grupo Antolin (Burgos, Spain) with reference GANF4 an average length of 5 µm, an average diameter of 45 nm, and a volume resistivity of 10^−4^ Ω.m (powder). The multi-walled carbon nanotubes (CNT) were supplied by Nanocyl (Sambreville, Belgium) with reference NC7000 and purity of 90%. The CNT had an average length of 1.5 μm and outer mean diameter of 9.5 nm. The graphene nanoplatelets were purchased from GRAFEN (Ankara, TURKEY) with reference Grafen iGP2, a diameter of 5 µm, a thickness between 5 and 8 nm, a surface area of 120–150 m^2^/g and an oxygen percentage below 1%.

The solvent used to disperse the fillers and dissolve the polymer was cyclopentyl methyl ether (CPME), supplied from Carlo Erba (Milan, Italy), with a density of 0.86 g/cm^3^ at 20 °C. This solvent provides a green solution by minimizing the solvent waste process, as CPME is a green solvent in comparison with toluene—the most-used solvent to process solvent-casted SEBS [18].

Figure 1 shows a schematic representation of the TPE, carbonaceous nanoparticles, and solvent used, together with the different steps for the preparation of nanocomposite films.

### 2.2. Sample Preparation

The preparation of the composites with 0, 0.25, 0.5, 1, 2, and 5 weight percentage (wt.%) of the three different carbonaceous nanofillers began by using the CPME solvent to homogeneously disperse each of them in an ultrasound bath (ATU-Ultrasonic, Model ATM40-3LCD, Valencia, Spain) for 3 h (step 1 in Figure 1), where the temperature changed between 25 and 30 °C. After filler dispersion, SEBS was added to the filler/solvent solution in a ratio of 1 g of SEBS to 6 mL of CPME. The solution was magnetically stirred (step 2) at 25 °C until complete dissolution of the SEBS polymer (about 3 h). 

After ultrasound and magnetic stirring, the nanocarbonaceous/SEBS composites were prepared by solution casting (step 3) on a glass substrate (previously cleaned with acetone), spreading by doctor blade technique with a controlled thickness of 750 μm, and drying for 12 h until total solvent evaporation at 20 °C. The final thickness of the films was near 50 µm after complete evaporation of solvent, with an area of 15 × 25 mm^2^. The composite nomenclature for the several nanofillers is presented in Table 1 for better understanding of the text. 

### 2.3. Nanocomposite Characterization

A Hitachi S-4800 field emission scanning electron microscope (FE-SEM, Lisboa, Portugal) at an accelerating voltage of 10 kV was used to assess the morphology of the samples with different nanofillers and nanofiller contents in the SEBS matrix. All samples were gold-coated (10 nm thick coating) in an Emitech K550X sputter coater (Quorum Technologies, Laughton, UK). 

Fourier-transform infrared spectroscopy (FTIR) analysis was performed in a Jasco FT/IR-4100 (Jasco, Porto, Portugal) in attenuated total reflectance (ATR) mode from 4000 to 600 cm^−1^, with 64 scans and a resolution of 4 cm^−1^.

Differential scanning calorimetry (DSC), METTLER DSC1 from METTLER TOLEDO (Barcelona, Spain) was used to perform thermal analysis of the composite samples. Two heating scans were performed for all samples. An initial heating scan was performed from −70 to 200 °C at 10 °C/min. After cooling to −70 °C at 10 °C/min, another heating scan was performed using the same parameters as for the first scan.

Thermogravimetric analysis (TGA), METTLER TGA/DSC1 (Barcelona, Spain) was conducted from 25 to 600 °C at a heating rate of 10 °C/min under nitrogen atmosphere. 

Mechanical measurements were carried out in a universal testing machine (Shimadzu model AG-IS, Izasa, Porto, Portugal), in tensile mode, with a load cell of 50 N. Rectangular samples with dimensions 15 × 8 mm^2^ and ≈50 µm thickness (measured with a digital micrometer Fischer Dualscope 603-478, Aveiro, Portugal) were analyzed at a test velocity of 5 mm/min.

The d.c. electrical resistance was evaluated from the I–V characteristic curves obtained by applying voltages between −10 V and +10 V and measuring the current, with an automated Keithley 487 picoammeter/voltage source (Solon, Ohio, USA). The volumetric electrical resistance of the films was calculated from the slope of the I–V curves, the measurements being collected from the previously deposited 5 mm diameter Au electrodes. The electrical conductivity was then calculated, taking into account the geometrical parameters of the samples.

Dielectric measurements (capacitance and the dielectric loss) were carried out at ≈25 °C in a frequency range of 100 Hz to 1 MHz, using an automatic Quadtech 1929 Precision LCR meter (Marlborough, MA, USA). The conductive electrodes had the same geometry as the ones used in the electrical resistivity measurements, leading to a parallel plate capacitor geometry. The dielectric constant was then obtained from the capacity measurements, taking into account the geometric characteristics of the samples.

The annealing treatment was carried out in a JP Selecta Digitronic oven (Series 2000, Barcelona, Spain) at 120 °C for 1, 24, and 72 h for SEBS and composites with 5 wt.% filler contents (CNF, CNT, and graphene). Overall properties of the samples were studied after annealing. 

Piezoresistive properties were measured using a Shimadzu AG-IS universal testing machine (50 N load cell, Izasa, Porto, Portugal) by applying uniaxial deformation cycles to the composite materials with simultaneous measurement of the electrical resistance through silver-painted electrodes (with conductive silver paint from Agar Scientific with reference AGG3790, Stansted, UK) connected to an Agilent 34401A multimeter (SOQUIMICA, Porto, Portugal). The electromechanical tests were performed at a velocity of 3 mm/min, varying the strain from 1% to 10%. The CNT/SEBS materials analyzed were pristine and aged (72 h at 120 °C) composites with 5 wt.% filler content. The electromechanical performance was evaluated by the gauge factor (GF) with contributions from the intrinsic piezoresistive effect and from geometrical factors given by:(1)$$GF=\frac{dR/R_{0}}{dl/l_{0}}=\frac{d\rho /\rho_{0}}{\varepsilon }+\left(1+2\nu \right),$$
where *R* is the electrical resistance, the relative deformation is given by $dl/l_{0}=\varepsilon$, *ρ* is the electrical resistivity, and *ν* is the Poisson ratio. The elastic properties of the SEBS and its nanocomposites influence the geometrical factor (1 + 2*ν*), and the maximum contribution is GF ≈ 2 for ideal elastomers with a *ν* = 0.5.

## 3. Results and Discussion

### 3.1. Morphological and Chemical Properties

The morphological analysis was carried out by SEM in SEBS and its composites with the different nanofillers with 5 wt.% filler content, as represented in Figure 2. Figure 2A shows the cross section of SEBS, showing the compact morphology typical of the polymer, with no porosity or voids. The composites in Figure 2B–D present a similar morphology, with a compact microstructure and homogeneous nanofiller dispersions that depended on the filler type. 

The dispersion of the different fillers within the SEBS matrix presented different features. CNF and G composites showed an individual particle dispersion in the composites, while the CNT composites presented a homogeneous agglomerate dispersion, with diameters of a few microns, as shown in the insets in Figure 2C. Considering that the processing method was the same for all composites, these varying nanofiller dispersions are fully ascribed to filler surface characteristics, determining the filler–matrix interaction, and to filler–filler interactions. Thus, CNT clusters were due to van der Walls forces between nanofillers [22], the TPE polymer not having a substantial role in the CNT distribution [11,20]. Note that homogeneous clusters of conductive nanofillers dispersed within the polymer matrix can improve the electric and piezoresistive properties with respect to individual nanofiller dispersion [11]. CNF and graphene fillers showed an individual nanoparticle dispersion pattern in the SEBS, as shown Figure 2B,D. On the other hand, the 3D distribution of nanoparticles in a polymer matrix can be different according to their geometry, their surface areas being drastically higher than their counterparts with sizes within the micrometer scale [22]. 

The FTIR spectra were obtained for neat SEBS and the several nanocomposites (Figure 3). Table 2 shows the characteristic bands for SEBS and its composites.

Figure 3A shows the FTIR spectra of neat SEBS and the SEBS composites with different contents of graphene, and Figure 3B shows the FTIR spectra of the different composites with 0.25 and 5 wt.% filler contents. 

The typical absorption bands of SEBS were observed at 2961, 2918, and 2851 cm^−1^ for the –C–H stretching of the aliphatic chain of the three blocks [18]. The band at 1601 cm^−1^ corresponds to the stretching of C=C bond from the aromatic ring of styrene, and the two bands related to –C–H bond bending were identified at 1491 and 1453 cm^−1^. The bands observed at 1377 and 1028 cm^−1^ correspond to the –C–CH_3_ bending from the methyl groups of butylene. Finally, the band at 755 cm^−1^ is related to the –C–H out-of-plane bending bond and the one at 699 cm^−1^ corresponds to the aromatic ring of the =C–H bond of the styrene block. All spectra were similar and independent of the nanofiller content (Figure 3A). As shown in Figure 3B for the SEBS composites with 0.25 and 5 wt.% filler contents, all spectra were similar for the three carbonaceous nanoparticles regardless of the nanofiller type and content, and the characteristic bands of SEBS remained unaltered after the addition of the different nanofillers, indicating no chemical interaction between the fillers and the polymer matrix.

### 3.2. Thermal Properties

Thermal measurements of SEBS and its nanocomposites with filler content up to 5 wt.% are presented in Figure 4A–D. DSC and TGA analyses were performed in order to study the influence of the incorporation of the different nanofillers on the thermal degradation and main transition temperatures.

The glass transition temperature (*T_g_*) of SEBS and its nanocomposites as a function of the CNT content up to 5 wt.% is shown in Figure 4A. SEBS has two distinct phases (i.e., ethylene-butylene and polystyrene [24]), but the SEBS thermogram was characterized by just one thermal event near 35 °C, related to the ethylene-butylene domains [10,25,26], indicating an amorphous polymer state. This transition temperature depends on the soft/hard phase ratio and on their block structure [11,27]. The molecular structure of SEBS, which includes two pure microphases of polystyrene and poly(ethylene-butylene) domains, influences its thermal behavior and the observed *T_g_* [27]. It is also evident from Figure 4A that the incorporation of 5 wt.% of carbonaceous nanofillers did not changes the *T_g_*, but it increased the mobile amorphous fraction involved in the transition, from greater to lesser increases of CNF, CNT, and G filler types, as estimated by the *ΔCp* values (i.e., the variation of heat capacity during glass transition) [28]. This fact is related to the different sizes and dimensionality of the nanofillers. Nanocomposites also showed no variations of *T_g_* as a function of the filler content (Figure 4B for CNT), indicating—as in the case of the FTIR spectra—no relevant chemical interactions between the fillers and the polymer matrix. SEBS composites with low nanocarbonaceous contents showed no changes in the glass transition temperature, as reported in the literature for related materials [11].

Thermogravimetric analyses of neat SEBS and the corresponding composites are presented in Figure 4C. The SEBS and composites presented an initial degradation temperature near 350 °C, with complete degradation around 470 °C, in agreement with the literature [29,30]. The inclusion of the fillers had no observable effect on these temperatures. The TGA thermograms were similar for SEBS and its nanocomposites with different filler contents, except for the residual mass, which increased with increasing filler concentration in the composites. The initial temperature was near 415 °C and the maximum temperature of degradation (DTG) was 450 °C. Figure 4D shows the residual weight (at 600 °C) and it agreed with the filler content in the composites (neat SEBS presented 0.4 wt.% residue and the nanocomposites showed a residue from 2.2 to 2.4 wt.%). The material weight loss at 450 °C (near DTG) was near 61 wt.% for SEBS, decreasing for composites up to 50.8 wt.% for composites with CNT as reinforcement material. Independent of the filler content, all nanocomposites showed a lower weight loss at 450 °C, indicating enhanced interfacial interactions between carbonaceous nanoparticles and the SEBS matrix. 

### 3.3. Mechanical and Electrical Properties

The mechanical behavior of the composites with different carbonaceous nanofillers is shown in Figure 5. The stress–strain curves were characterized first by a linear variation between stress and strain, where the materials’ mechanical resistance to tensile stress was mainly dominated by the soft part of the copolymer (ethylene/butylene) [27,31] until the yield stress and strain of the polymer matrix was reached. Then, after yielding, the elastomer characteristics were shown in the strain, which continued increasing up to >500% for SEBS for stresses up to near 50 MPa. Thus, SEBS and their composites showed a typical behavior of thermoplastic elastomers [32]. Pristine SEBS showed a larger maximum stress than the composites due to the lack of chemical interaction between fillers and the polymeric matrix. Fillers can thus be considered as defects within the polymer structure and their effects on the mechanical properties depend on agglomerate size and distribution.

The maximum strains of the materials varied from 450% to 600% in nanocomposites reinforced with CNF and CNT (Figure 5A,B). In the case of G (Figure 5C), the maximum strain decreased with increasing graphene content, down to 150% strain for the 5G/SEBS composite.

G/SEBS composites showed a decrease in the maximum strain for larger filler contents (i.e., higher than 1 wt.%), decreasing from 450% to 150% for composites with 1 and 5 wt.% of graphene, respectively. This decrease of the deformation capability of the SEBS composites with larger nanofiller contents is typically ascribed to graphene agglomerates [33] (which were not observed in our case), or to graphene interactions with polymeric chains—mainly polystyrene (PS) domains [34]. It has been reported that the distance between nanodomains of styrene and ethylene/butylene increases with increasing graphene content in nanocomposites [34]. When analyzed by small-angle X-ray scattering (SAXS) and atomic force microscopy (AFM), nanocomposites with larger graphene contents showed significant distortions from a hexagonal arrangement [34]. The initial modulus of the SEBS and composites with different nanocarbonaceous fillers (Figure 5D) was around 80 MPa for SEBS and was practically constant, independent of the filler content up to 5 wt.%. 

The electrical and dielectric behavior of the SEBS composites with different fillers and filler contents (up to 5 wt.%) is presented in Figure 6. The electrical behavior of the composites depended critically on the filler type (Figure 6A). CNT/SEBS composites showed percolation thresholds (PTs) between 1 and 2 wt.% of CNT, while the other nanofillers (i.e., CNF and graphene) did not present PTs for up to 5 wt.% nanofiller contents. 

The electrical properties of carbonaceous nanocomposites are critical for the overall performance of the materials for sensor applications. The intrinsic electrical properties and geometrical characteristics of the different fillers influence the conductivity of the composites. CNT/SEBS composites present distinct electric and dielectric properties compared to G and CNF composites. Although CNT and CNF show high aspect ratio compared to G sheets, the greater length (crossing the styrene and ethylene/butylene nanodomain sizes [34]) of the CNF does not allow the electrical conductivity of the composites to be improved. CNF and G showed a reduced intrinsic electrical conductivity, and the composites were insulators up to 5 wt.% of filler incorporation. The electrical conductivity, dielectric constant, and losses (Figure 6B) of the composites with CNF and G showed comparable dielectric properties up to 5 wt.% of nanofiller content. The dielectric constant was equivalent for G and CNF, except for the composites with larger filler contents. The dielectric constant of SEBS was near *ε*’≈ 2 and increased up to *ε*’≈ 2.5, 5, and 8 for CNT/SEBS (1 wt.%), G/SEBS, and CNF/SEBS (5 wt.%), respectively. For all used nanofillers, the dielectric losses increased with increasing filler content in the composite with respect to the pristine SEBS. Due to the intrinsic electrical conductivity of CNT, the dielectric losses of the nanocomposites were higher compared to those of G and CNF, even just for 1 wt.%. The dielectric losses were higher for CNF than G nanocomposites, and the difference increased with increasing filler content in the composite. G and CNF can be used for the preparation of highly dielectric materials, but CNF presented a slightly higher electrical conductivity (Figure 6A) and the dielectric losses were also larger. CNT/SEBS composites showed different dielectric behavior than the other nanofillers, with nearly constant dielectric response up to 1 wt.%, becoming conductive after this CNT content, as observed by the electrical conductivity values. Therefore, CNT are the ideal reinforcement material for the development of conductive and piezoresistive polymer-based composites, and graphene and CNF appear to be more appropriate for improved dielectric materials.

### 3.4. Thermal Annealing

The SEBS and corresponding composites with 5 wt.% were submitted to thermal treatment at 120 °C for 1, 24, and 72 h in order to evaluate eventual physico-chemical modifications and/or variations of the functional response after the materials reached temperatures of 120 °C, leading to limitations to the use of the materials for applications. Thermal and photooxidation can deteriorate the overall properties of the SEBS and composites [35]. FTIR and DSC analyses revealed that the annealing process did not influence the chemical or thermal properties of the materials (Figure 7A,B, respectively). 

Figure 7A shows the typical absorption bands of the SEBS and composites with 5 wt.% filler content for samples without and with thermal annealing. It was observed that the annealing absorption band near 1720 cm^−1^ (filled blue line in the Figure 7A)—a peak corresponding to the aliphatic aldehydes (corresponding to C=O stretch) related to the oxidation of the ethylene/butylene block—does not appear for SEBS with 72 h at 120 °C, as reported in the literature [23,36]. Thus, the chemical stability of SEBS and corresponding composites was demonstrated, and the materials did not show any degradation up to temperatures of 120 °C, in contrast with some reports in the literature [36].

Thermal analyses showed that the glass transition temperatures (Figure 7B) were similar for SEBS and composites (as previously demonstrated), but also similar for the composites with different carbon nanofillers with annealing times up to 72 h. The thermograms indicate a T_g_ ≈ 35 °C for SEBS and CNF/SEBS composites with 1, 24, and 72 h annealing time at 120 °C. There were also no changes to report after the annealing of other composites.

Mechanical and electrical properties after annealing (120 °C for 72 h) for SEBS and composites with 5 wt.% of carbonaceous nanofillers are presented in Figure 8.

The mechanical and electrical properties of the SEBS and the corresponding nanocomposites are presented in Figure 8A,B, respectively. The mechanical properties of the annealed materials were similar to those of the non-annealed ones, with similar maximum strain levels, but with lower maximum stress, mainly for the nanocomposites. Aged G/SEBS composites showed maximum strain near 300%, and the remaining materials showed maximum strains larger than 500%. In this way, aged materials also showed excellent properties for large-strain applications. 

However, the electrical conductivity decreased slightly with ageing time for all materials (SEBS and composites with 5 wt.%), by around one order of magnitude with respect to pristine samples for 72 h at 120 °C of ageing. CNT/SEBS composites showed stable electrical conductivity for aged samples (constant electrical conductivity with increasing annealing time) and G/SEBS composites were the material with the largest decrease in the electrical conductivity. This decrease was related to the behavior of the SEBS matrix (also showing such reduction of the electrical conductivity), and therefore with polymer–filler interface effects and reconfiguration of the nanofiller network. 

### 3.5. Electromechanical Properties

Polymer-based composites can be tailored as sensors materials, and thermoplastic elastomer or elastomer composites show excellent properties for large-strain sensor materials due to their mechanical properties. SEBS reinforced with CNT composites (with 5 wt.% fillers) presented linear electromechanical behavior (Figure 9A) and good performance for sensors without and with ageing, as can be seen in Figure 9B. Note that the electrical conductivities of the remaining composites studied in the present work were too low for electromechanical sensing applications [11,17]. 

Polymer-based composites with appropriate electrical conductivity can show electrical resistance variation proportional to the mechanical stimulus applied to the composite, increasing the resistance while the samples are stressed and decreasing when sample recovers to the initial deformation (Figure 9A). Nevertheless, there was an initial reduction of the electrical resistance with the number of loading cycles, suggesting a rearrangement of the conductive network with deformation—a typical electromechanical behavior of elastomeric polymer composites [16,31,37]. It has been shown that after some aging cycles the mechanical and electrical signals tend to stabilize, leading to materials with a stable electromechanical response that is suitable for applications [18]. 

The CNT/SEBS composite with 5 wt.% filler content presented excellent electromechanical behavior, with a GF near 2 for the as-prepared material, decreasing from GF ≈ 0.5 to 1, depending on the applied strain, for samples with 72 h at 120 °C of annealing, as can be observed in Figure 9B. Thus, the variation of electrical resistance followed the external stimulus applied to the sensor, presenting a high linearity between both measurements. Thus, CNT/SEBS composites can work as large-strain sensors (10% at least) with good linearity and performance for pristine samples. The aged sensor also showed good linearity with a lower sensibility performance (0.5 < GF < 1). 

## 4. Conclusions

Thermoplastic elastomer SEBS composites with CNF, CNT, and graphene nanofillers were prepared and characterized. It was shown that the excellent mechanical properties of SEBS were maintained nearly constant, independent of the reinforcement materials. The maximum strain and stress at rupture could reach strains on the order of 600%. It was observed that all fillers were properly dispersed for graphene and CNF. Small agglomerates were nevertheless observed for CNT fillers within the polymer SEBS matrix. Further, thermal and chemical properties of the composites did not show strong variations with filler type and content. 

Electrical properties of the composites depended strongly on the intrinsic properties of the fillers, for the same processing method. CNT/SEBS composites presented percolation thresholds lower than 2 wt.% CNT, while CNF and graphene composites did not show percolation up to 5 wt.%. The dielectric response of the composites was better for CNF/SEBS composite with 5 wt.% filler, reaching a dielectric constant of 8—nearly four times that of pristine SEBS. The piezoresistive response was evaluated in CNT/SEBS composites, showing a GF > 2 and good linear response for deformations up to 10% for the composite with 5 wt.% filler content. Annealing of the materials did not lead to relevant physical-chemical modifications, but the GF was reduced to ≈1 after annealing for 72 h at 120 °C, showing the applicability of the materials for sensing applications.

## Figures and Tables

**Figure 1 materials-12-01405-f001:**
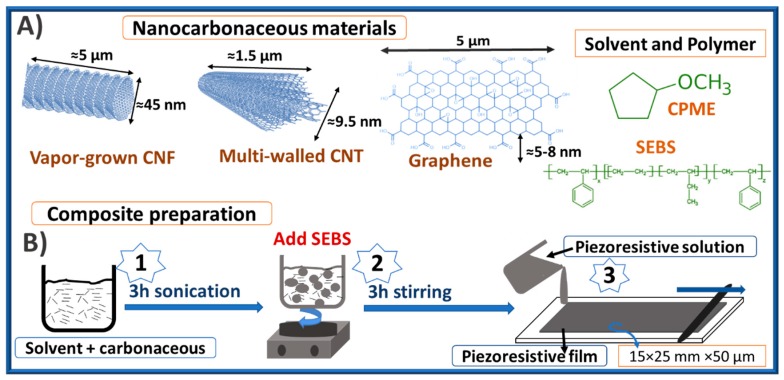
Schematic representation of (**A**) the used nanocarbonaceous materials, solvent, and polymer; and (**B**) the composite preparation route. CNF: carbon nanofibers; CNT: carbon nanotubes; CPME: cyclopentyl methyl ether; SEBS: styrene-b-(ethylene-co-butylene)-b-styrene.

**Figure 2 materials-12-01405-f002:**
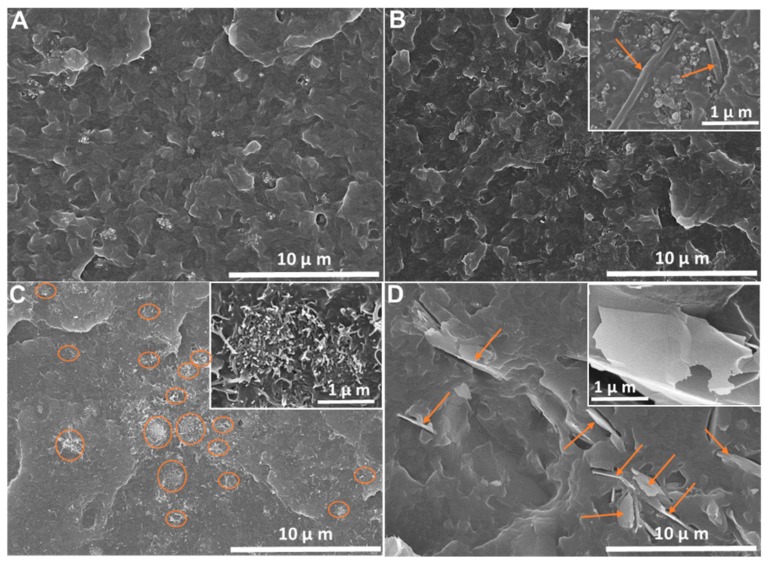
SEM images of (**A**) SEBS and its composites with 5 wt.% of (**B**) CNF, (**C**) CNT, and (**D**) graphene at 5000× magnification. The magnification of the insets in (B–D) is 50,000×.

**Figure 3 materials-12-01405-f003:**
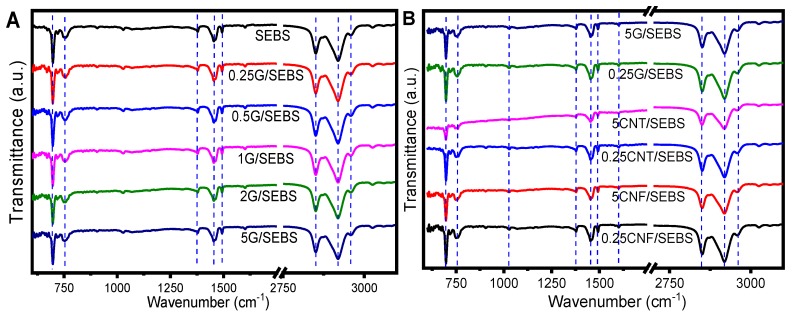
FTIR spectra of SEBS and its nanocomposites as a function of (**A**) the graphene (G) content up to 5 wt.% and (**B**) for the different composites with CNF, CNT, and G for filler contents of 0.25 and 5 wt.%.

**Figure 4 materials-12-01405-f004:**
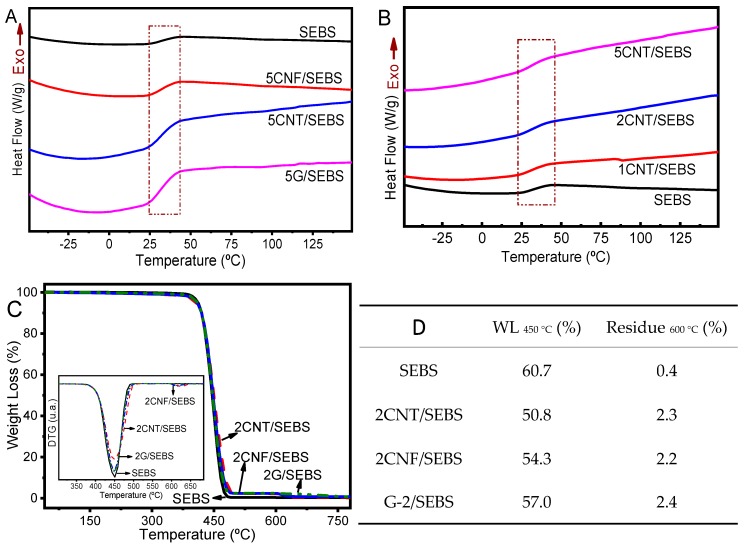
(**A**,**B**) DSC thermograms for SEBS and its composites with different carbonaceous nanofillers; (**C**) TGA and DTG of the SEBS and its nanocomposites with 2 wt.% of the different fillers; (**D**) Thermal degradation of the materials based on the TGA measurements. DTG: derivative thermogravimetry; WL: weight loss.

**Figure 5 materials-12-01405-f005:**
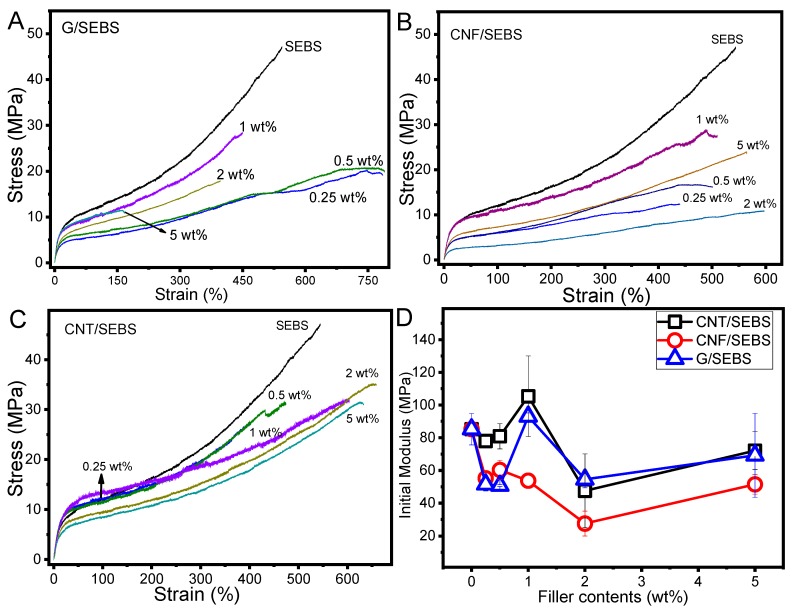
Mechanical properties of SEBS and its nanocomposites reinforced with (**A**) CNF; (**B**) CNT; and (**C**) graphene up to 5 wt.%. (**D**) The initial modulus of the materials as a function of nanofiller content.

**Figure 6 materials-12-01405-f006:**
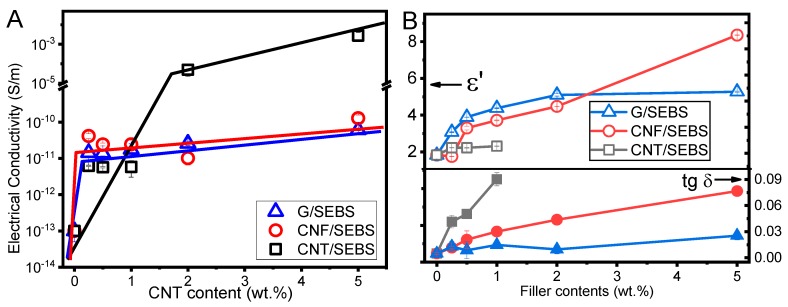
(**A**) Electrical properties of the SEBS and composites with G, CNF, and CNT. The lines are for guiding the eyes; (**B**) Dielectric constant and losses for SEBS composites with different nanofillers at 1 kHz.

**Figure 7 materials-12-01405-f007:**
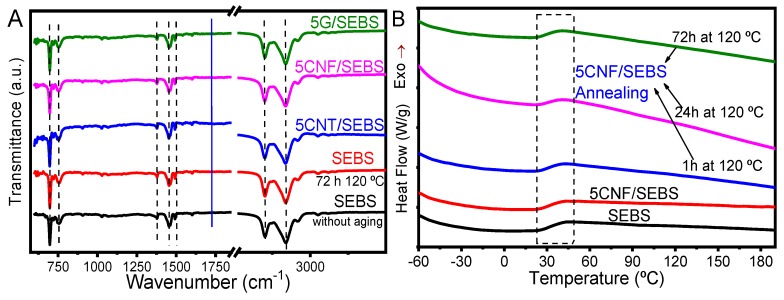
Annealed samples measured by (**A**) FTIR and (**B**) DSC. Annealed SEBS and respective composites with 5 wt.% at different annealing times from 1 to 72 h, at 120 °C.

**Figure 8 materials-12-01405-f008:**
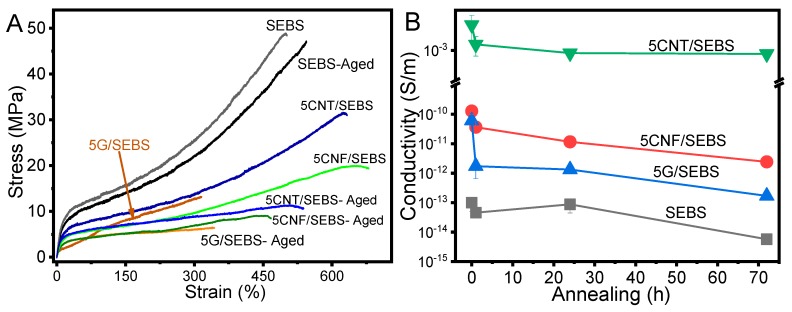
(**A**) Mechanical properties of SEBS and its composites with 5 wt.% of different fillers (graphene, CNF, and CNT) at 5 mm/min before and after annealing for 72 h at 120 °C; (**B**) Electrical conductivity for SEBS and respective nanocomposites with 5 wt.% filler content for the different annealing times at 120 °C.

**Figure 9 materials-12-01405-f009:**
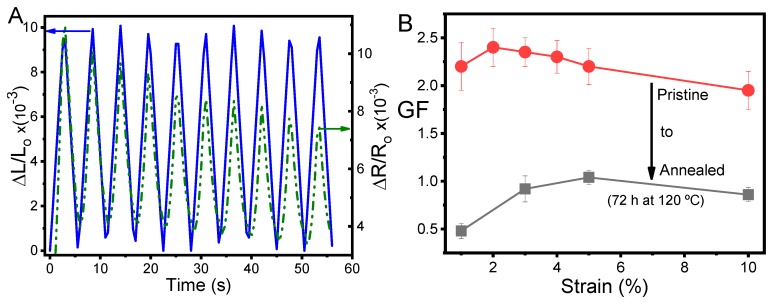
Electromechanical performance of pristine and annealed (72 h at 120 °C) 5CNT/SEBS composite. (**A**) Electromechanical behavior for 1% strain at 3 mm/min for 10 loading–unloading cycles; (**B**) Electromechanical sensibility of the pristine and annealed sensing materials measured at 3 mm/min from 1% to 10% strain. GF: gauge factor.

**Table 1 materials-12-01405-t001:** Nomenclature of the SEBS composites with different nanocarbonaceous fillers and contents.

Filler Content	Nomenclature		
	CNF/SEBS	CNT/SEBS	Graphene/SEBS
0.25 wt.%	0.25 CNF/SEBS	0.25 CNT/SEBS	0.25 G/SEBS
0.5 wt.%	0.5 CNF/SEBS	0.5 CNT/SEBS	0.5 G/SEBS
1 wt.%	1 CNF/SEBS	1 CNT/SEBS	1 G/SEBS
2 wt.%	2 CNF/SEBS	2 CNT/SEBS	2 G/SEBS
5 wt.%	5 CNF/SEBS	5 CNT/SEBS	5 G/SEBS

**Table 2 materials-12-01405-t002:** Characteristic FTIR vibration bands of SEBS.

Wavenumber (cm^−1^)	Vibrational Mode	Ref
699 755	=C–H bending –C–H– out of plane bending	[18,23]
1028 1377	–C–CH_3_ bending	
1453 1491	–C–H bending	
1601	C=C stretching	
2851 2918 2961	–C–H stretching	

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
