# Peer review of "Carbonaceous Filler Type and Content Dependence of the Physical-Chemical and Electromechanical Properties of Thermoplastic Elastomer Polymer Composites"

_materials, 2019, doi:10.3390/ma12091405_

Round 1

Reviewer 1 Report

- Abstract is too long. For example, reducing the first paragraph to 1 or 2 sentences is necessary.

- Putting “2. Results and discussion” ahead of “3. Materials and Methods” is abnormal and strange. The order should be reversed.

- Detailed information (mol. wt., PDI, density, etc.) on the SEBS is necessary.

- Was the solution casting process stable enough to get the films with even distribution of the fillers and thickness?

- What is the purpose of the FTIR analysis? The FTIR spectra does not give any meaningful information or results.

- According to the DSC curves the nanocomposites show no variations of Tg as a function of the nanofiller type and content. However, the carbon nanofillers generally increase Tg of a polymer matrix.

- Figure 4: Was the same solution casting process used to prepare the neat SEBS film? How many specimens were made and tested for each sample? It is abnormal looking the S-S curve of the neat SEBS in the upper area of the graph.

- Figure 6 does not give any meaningful information or results.

- What is the purpose of the thermal annealing? It looks the thermal annealing induced just thermal degradation resulting in the physical properties decrease.

Author Response

Abstract is too long. For example, reducing the first paragraph to 1 or 2 sentences is necessary.

A: The abstract has been reduced in 4-5 lines.

- Putting “2. Results and discussion” ahead of “3. Materials and Methods” is abnormal and strange. The order should be reversed.

A: We followed the author guidelines, indicating:  Introduction, Results, Discussion, Materials and Methods, Conclusions (optional). Nevertheless, we agree with the referee and have modified the paper as indicated.

- Detailed information (mol. wt., PDI, density, etc.) on the SEBS is necessary.

A: All available information in the materials datasheet has been included. - Was the solution casting process stable enough to get the films with even distribution of the fillers and thickness?

A: Yes. Solution casting with tailored viscosity and specific solvent evaporation conditions allow the processing of polymeric composites films with even distribution of fillers (better as by any other method) and thickess.

- What is the purpose of the FTIR analysis? The FTIR spectra does not give any meaningful information or results.

A: FTIR provides information on the characteristic’s peaks of the polymer, which may suffer variations due to filler-polymer interactions. In the present case, the polymer characteristic bands remain unchanged, independently of the filler content.

- According to the DSC curves the nanocomposites show no variations of Tg as a function of the nanofiller type and content. However, the carbon nanofillers generally increase Tg of a polymer matrix.

A: It depends on the polymer, fillers and filler-polymer interactions. As we observe by FTIR analysis, fillers and SEBS matrix does not show any chemical interaction. Thus, Tg remains practically similar to pristine polymer.

- Figure 4: Was the same solution casting process used to prepare the neat SEBS film? How many specimens were made and tested for each sample? It is abnormal looking the S-S curve of the neat SEBS in the upper area of the graph.

A: Yes, all samples were made using the same method. Three specimens were testes for each sample. The S-S curve is the characteristic one of SEBS and does not suffer significant variations due to the low filler contents.  

- Figure 6 does not give any meaningful information or results.

A: Figure 6 shows the characteristics FTIR and DSC signatures of the polymer and composites, showing the lack of significant influence of the vibration bands and on the Tg. This information is very valuable and needed to properly understand the effect of the fillers on the polymer matrix.

- What is the purpose of the thermal annealing? It looks the thermal annealing induced just thermal degradation resulting in the physical properties decrease.

A: Thermal annealing, on the one hand, may influence polymer and filler stability and interaction. On the other hand, it allows to evaluate whether the material can be used as piezoresistive sensors up to that temperatures.

Reviewer 2 Report

The submitted manuscript investigates electromechanical, morphology, and physical-chemical properties of SEBS composites containing Graphene, CNT, and CNF nanofillers at various concentrations. Some of the interesting parts of the manuscript are the percolation threshold presented in Figure 5 and the piezoresistive response of CNT/SEBS composites.

Although a great deal of researches can be found in the literature, the reviewer is in favor of publishing this work in Materials journal. The manuscript has been well-written, and the results have been clearly presented.

There are just a couple of minor remarks that need to be addressed prior to publication.

The authors could comment in more details on the TGA results (e.g. a more comprehensive explanation on lower weight loss at 450 °C for nanocomposites compared to neat SEBS, as well as the reason for remarkable weight loss differences between Graphene and CNT composites).

Please provide some discussion for Figure 4D (Modulus vs filler concentration) in the text.

In the Conclusion section, line 407, it is mentioned that “… all fillers are properly dispersed within the polymer SEBS matrix …”. This conclusion needs to be revised as the SEM image in Figure 1C exhibits CNT agglomerates.

Author Response

The submitted manuscript investigates electromechanical, morphology, and physical-chemical properties of SEBS composites containing Graphene, CNT, and CNF nanofillers at various concentrations. Some of the interesting parts of the manuscript are the percolation threshold presented in Figure 5 and the piezoresistive response of CNT/SEBS composites.

Although a great deal of researches can be found in the literature, the reviewer is in favor of publishing this work in Materials journal. The manuscript has been well-written, and the results have been clearly presented.

There are just a couple of minor remarks that need to be addressed prior to publication.

The authors could comment in more details on the TGA results (e.g. a more comprehensive explanation on lower weight loss at 450 °C for nanocomposites compared to neat SEBS, as well as the reason for remarkable weight loss differences between Graphene and CNT composites).

It has been added: The SEBS and composites present initial degradation temperature near 350 ºC, with complete degradation around 470 ºC, in agreement with the literature [29, 30]. No relevant effect of those temperatures is observed with the inclusion of the fillers.

Please provide some discussion for Figure 4D (Modulus vs filler concentration) in the text.

It has been added:The initial modulus of the SEBS and composites with different nanocarbonaceous fillers (Figure 4D) is around 80 MPa for SEBS and is practically constant independently of the filler content up to 5 wt.%.  

In the Conclusion section, line 407, it is mentioned that “… all fillers are properly dispersed within the polymer SEBS matrix …”. This conclusion needs to be revised as the SEM image in Figure 1C exhibits CNT agglomerates.

It has been added: It is observed that all fillers are properly dispersed, for graphene and nanofibers. Small agglomerates are observed for CNT within the SEBS polymer.

Reviewer 3 Report

This manuscript presented improved mechanical properties, electrical properties, thermal and chemical response, dielectric response of thermoplastic elastomer SEBS composites with CNF, CNT and graphene. Authors have submitted a decent manuscript for publication. The overall presentation of their efforts is good. They have described in details their methods and analysed extensively their results. I am happy to suggest the acceptance of this work for publication after some major and minor changes.

Major changes:

1.       Material and methods section should be moved before the results and discussion part.

Minor changes:

1.       Reviewer suggest some minor changes in title, physical-chemical should be changed to “physicochemical” and propertied should be changed to “properties”.

2.       Line 31: larger than should be replaced with “above”

3.       Line 133: Figure 1C should be used instead of Figure 2

4.       Line 140: Figures 1B and D can be written as Figures 1B and 1D  

5.       Line 166: Figure 3A to 4D should be changed to Figure 3A to 3D

6.       In whole manuscript °C should be used in place of ºC.

7.       Line 217 : Graphene is to be used in place of grapheme

8.       In figure 5 uniform font size of legends to be used.

9.       Line 254: for 1 h, 24 h and 72 h is used in place of for 1, 24 and 72 h

10.   Line 334 : g/cm3 is to be used.

11.   In equation (1) uniform font size to be used

Author Response

This manuscript presented improved mechanical properties, electrical properties, thermal and chemical response, dielectric response of thermoplastic elastomer SEBS composites with CNF, CNT and graphene. Authors have submitted a decent manuscript for publication. The overall presentation of their efforts is good. They have described in details their methods and analysed extensively their results. I am happy to suggest the acceptance of this work for publication after some major and minor changes.

Major changes:

1.       Material and methods section should be moved before the results and discussion part.

A: It has been moved, as requested.

Minor changes:

1.       Reviewer suggest some minor changes in title, physical-chemical should be changed to “physicochemical” and propertied should be changed to “properties”.

2.       Line 31: larger than should be replaced with “above”. Corrected

3.       Line 133: Figure 1C should be used instead of Figure 2. Corrected

4.       Line 140: Figures 1B and D can be written as Figures 1B and 1D. Corrected 

5.       Line 166: Figure 3A to 4D should be changed to Figure 3A to 3D. Corrected

6.       In whole manuscript °C should be used in place of ºC. Corrected

7.       Line 217 : Graphene is to be used in place of grapheme. Corrected

8.       In figure 5 uniform font size of legends to be used. Corrected

9.       Line 254: for 1 h, 24 h and 72 h is used in place of for 1, 24 and 72 h. Corrected

10.   Line 334 : g/cm3 is to be used. Corrected

11.   In equation (1) uniform font size to be used. Corrected

Round 2

Reviewer 1 Report

- Detailed information (mol. wt., PDI) on the SEBS is still not included. The supplier would have the information.

- The FTIR spectra does not give any meaningful information because in the present case using neat carbon nanofillers, it can be easily predicted that the polymer characteristic bands remain unchanged, independently of the filler content. Likewise, Figure 7 in the revision does not give any meaningful information.

- Even though there is no chemical interaction between the fillers and the matrix, the carbon nanofillers generally increase Tg of the polymer matrix because of the confinement effect if the nanocomposites are well prepared.

- Figure 5 in the revision: It is abnormal looking the S-S curve of the neat SEBS in the upper area of the graph. It seems like that the nanocomposites are not well prepared.

- The purpose of the thermal annealing experiment should be included in the main text.

Author Response

- Detailed information (mol. wt., PDI) on the SEBS is still not included. The supplier would have the information.

A: We have added all available information, including also viscosity and hardness.

SEBS 6110 have a toluene solution viscosity of 470 cP and hardness of 76 (Shore A).

- The FTIR spectra does not give any meaningful information because in the present case using neat carbon nanofillers, it can be easily predicted that the polymer characteristic bands remain unchanged, independently of the filler content. Likewise, Figure 7 in the revision does not give any meaningful information.

A: We are glad that the referee can easily predict the filler polymer interactions and that thermal annealing does not cause any modification of the bands and thermal properties. As most of the researchers (see the many contributions in the field) we have to measure and demonstrate it. This is what we have performed and what is important to be reported, as it is reported in the large variety of works in polymer matrix composites, both with respect to the FTIR and, in particular, with the DSC results.

We have changed the paragraph in manuscript to:

Figure 7A shows the typical absorption bands of the SEBS and composites with 5 wt.% filler content for samples without and with thermal annealing. It is observed that the annealing absorption band near 1720 cm-1 (filled blue line in the Figure 7A), a peak corresponding to the aliphatic aldehydes (corresponding to C=O stretch) related with oxidation of the ethylene/butylene block, does not appear for SEBS with 72 h at 120 °C, as reported in the literature [23, 36]. Thus, the chemical stability of SEBS and corresponding composites is evidenced and the materials do not show any degradation up to temperatures of 120 °C, in contrast with some reports in the literature [36].  

- Even though there is no chemical interaction between the fillers and the matrix, the carbon nanofillers generally increase Tg of the polymer matrix because of the confinement effect if the nanocomposites are well prepared.

A.  This is not always true, as demonstrated extensively by the literature in the area, as depends on filler type and content, among other variables.

The manuscript was changed for:

SEBS composites with low nanocarbonaceous contents  show no changes in the glass transition temperature, as reported in the literature for related materials [11].

- Figure 5 in the revision: It is abnormal looking the S-S curve of the neat SEBS in the upper area of the graph. It seems like that the nanocomposites are not well prepared.

It seems like that the nanocomposites are not well prepared”. We would like more specification from the referee on this unfear and offensive statement.

It is not “abnormal” due to the lack of chemical interaction between the fillers and polymer, as demonstrated in the manuscript. Which is “abnormal” is to doubt on the preparation of the samples without reasons or evidence and after all the experimental results reported in the work.

Pristine SEBS shows a larger maximum stress than the composites due to the lack of chemical interaction between fillers and polymeric matrix. Fillers can thus be considered as defects within the polymer structure and their effect on the mechanical properties depend on agglomerate size and distribution.

- The purpose of the thermal annealing experiment should be included in the main text.

It has been included, please see p 10: “thermal treatments at 120 °C for 1 h, 24 h and 72 h in order to evaluate eventual physico-chemical modifications and/or variations of the functional response after the material reaching temperatures of 120 ºC, leading to limitations of the used of the materials for applications”   
